

# Modeling volcanic ash aggregation processes and related impacts on the April/May 2010 eruptions of Eyjafjallajökull Volcano with WRF-Chem.

Sean D. Egan[1], Martin Stuefer[2], Peter W. Webley[2], Taryn Lopez[2], Catherine F. Cahill[2], Marcus Hirtl[3]

[1]Department of Chemistry, University of Alaska Fairbanks, Fairbanks, AK, 99775, USA
[2]Geophysical Institute, University of Alaska Fairbanks, Fairbanks, AK 99775, USA
[3]Department of Chemical Weather Forecasting, Zentralanstalt für Meteorologie und Geodynamik (ZAMG), Wien, 1190, AT

*Correspondence to*: Sean D. Egan (sdegan@alaska.edu, sean.d.egan@navy.mil)

**Abstract.** Volcanic eruptions eject ash and gases into the atmosphere that can contribute to significant hazards to aviation, public and environment health, and the economy. Several volcanic ash transport and dispersion (VATD) models are in use to simulate volcanic ash transport operationally, but none include a treatment of volcanic ash aggregation processes. Volcanic ash aggregation can greatly reduce the atmospheric budget, dispersion and lifetime of ash particles and therefore its impacts. To enhance our understanding and modeling capabilities of the ash aggregation process, a volcanic ash aggregation scheme was integrated into the Weather Research Forecasting with online Chemistry (WRF-Chem) model. Aggregation rates and ash mass loss in this modified code are calculated in-line with the meteorological conditions, providing a fully coupled treatment of aggregation processes. The updated-model results were compared to field measurements of tephra fallout and in situ airborne measurements of ash particles from the April/May 2010 eruptions of Eyjafjallajökull Volcano, Iceland. WRF-Chem, coupled with the newly added aggregation code, modeled ash clouds that agreed spatially and temporally with these in situ and field measurements. A sensitivity study provided insights into the mechanics of the aggregation code by analyzing each aggregation process (collision kernel) independently, as well as by varying the fractal dimension of the newly formed aggregates. In addition, the airborne lifetime (e-folding) of total domain ash mass was analyzed for a range of fractal dimension, and a maximum reduction of 79.5% of the airborne ash lifetime was noted.

## 1. Introduction

Volcanic eruptions inject gases and ash particles of various sizes into the atmosphere, posing hazards to life, infrastructure and aviation (Miller and Casadevall, 2000). Volcanic emissions can alter the composition of the atmosphere and affect the Earth's radiation budget and climate (Angell, 1993; Cole-Dai, 2010; Thordarson and Self, 2003). The environmental and economic impacts of past and recent eruptions have spurred increased interest in the inclusion of volcanic ash into numerical weather prediction (NWP) models (Folch et al., 2009, 2015; Lin et al., 2012; Stuefer et al., 2013). Today, forecasters and scientists utilize volcanic ash transport and dispersion (VATD) models for ash hazard mitigation, the



development, calibration and validation of remote sensing tools, the study of ash physics. Numerical models have been developed to better describe the initial plume characteristics of eruptions. A current limitation of most VATD models is their ability to capture volcanic ash aggregation.

Volcanic ash aggregation is important for many reasons. Aggregation affects the atmospheric lifetime of ash, the distance ash is transported from the eruption source, the size and type of tephra observed on the ground, and the duration ash

poses a threat to aircraft (Brown et al., 2012; Casadevall, 1994; Rose and Durant, 2011). Aggregation has been observed in several well studied volcanic eruptions such as those of Mount St. Helens (Washington), Mount Redoubt (Alaska) and Eyjafjallajökull (Iceland). Additionally, aggregation occurs in both proximal (< 15 km from the plume corner) and distal ash clouds (Bonadonna et al., 2011; Bonadonna and Phillips, 2013; Brown et al., 2012; Carey and Sigurdsson, 1982; Rose and Durant, 2009, 2011; Taddeucci et al., 2011; Wallace et al., 2013).

Proximal volcanic ash aggregates form more rapidly than distal aggregates for a number of reasons. For example, ice and liquid water enhance the sticking of particles and thus increases the rate of aggregation (Brown et al., 2012; Rose and Durant, 2011). This process can occur in a hail-like process with a cycle of freezing and thawing leading to enhanced aggregation (Van Eaton et al., 2015). In addition, the higher concentration of ash in the proximal plume increases the number of collisions.

Water enhanced aggregation in the proximal plume has been observed in a number of eruptions. Field observations of tephra from the May 18, 1980 eruption of Mount St. Helens detail the formation of large volcanic aggregates (up to 1mm) closely correlated with the presence of rain, snow, and hail (Waitt et al., 1981). Gilbert and Lane (1994) note that aggregation rates were enhanced by high proximal water vapor concentrations during the eruptions of Sakurajima volcano in the 1990s, and the majority of this water-enhanced aggregation occurred proximally, within the first minutes of the eruption.

In addition, studies of the 2009 eruption of Mount Redoubt in Alaska show definitive evidence for aggregation enhanced sedimentation in the proximal plume (Van Eaton et al., 2015; Wallace et al., 2013). Van Eaton et al. (2015) conclude that the effects of aggregation in the Redoubt eruption resulted in over 95% of fine ash mass deposited to the ground as aggregates.

Distal aggregation usually occurs at a slower rate than proximal aggregation as the plume ages and diffuses (Rose and Durant, 2009, 2011). Despite a slower rate of aggregation the majority of distal fine ash settles to the ground as larger

aggregates (Brown et al., 2012; Carey and Sigurdsson, 1982; Rose and Durant, 2011; Wallace et al., 2013). Both coarse and fine ash particles are known to aggregate in distal clouds by forming dry clusters due to electrostatic attraction, or as liquid or frozen water particles (Brown et al., 2012; Rose and Durant, 2011). Distal aggregate formation has been observed from eruptions such as Etna Volcano, Italy in 1971, Mount St. Helens, U.S. in 1980 and Mount Redoubt, U.S. in 1990 (Booth and Walker, 1973; Sorem, 1982; Sparks et al., 1997). For many eruptions, electrostatic aggregation of fine ash is expected to be

responsible for the bimodal distribution of volcanic ash fallout (Carey and Sigurdsson, 1982; Cornell et al., 1983; James et al., 2003).

Recently, aggregation processes were observed to play an integral role in the dispersion of the plume generated from the April and May 2010 eruptions of Eyjafjallajökull Volcano, Iceland. In-situ measurements of ash particle fall



velocities using high speed photography observed aggregation-enhanced sedimentation that increased fallout rates by a
factor of 10 (Taddeucci et al., 2011). The effect of ash aggregation caused a significant quantity of additional ash fall across
Iceland, rather than be transported further. Ash aggregation overall clearly reduced the atmospheric residency time of the
Eyjafjallajökull ash plume (Gudmundsson et al., 2012). In addition, aggregation was observed to cause enhanced fallout over
parts of mainland Europe and the United Kingdom (Stevenson et al., 2012).

Aggregation processes not only affect the lifetime of volcanic ash, but also the makeup of volcanic ash cloud
particle size distributions (PSDs) which may complicate modeling and remote sensing efforts (Brown et al., 2012; Rose and
Durant, 2011). For example, volcanic ash remote sensing algorithms require information regarding particle sizes and
extinction coefficients (Stohl et al., 2011; Wallace et al., 2013). Remote sensing methods are also used to estimate eruption
parameters and PSDs via extinction coefficients using inverse modeling (Kristiansen et al., 2012; Stohl et al., 2011).
Additionally, volcanic PSDs are also important for the study of radiative properties of volcanic ash and their effects on the
atmosphere (Hirtl et al., 2019; Young et al., 2012).

The effects imposed on volcanic ash clouds by aggregation processes necessitates their parameterization in volcanic
ash transport and dispersion (VATD) models. Despite this, only few of the existing VATD models capture aggregation
processes. For example, a volcanic ash aggregation parameterization scheme has been implemented within the FALL3D
model (Folch et al., 2009). In an operational setting, FALL3D runs by ingesting offline meteorological fields from gridded
atmospheric models, such as the Weather Research Forecasting (WRF) model, and then calculating volcanic ash advection
and sedimentation during the parent model output time step. Another method of capturing volcanic ash aggregation is to
initialize VATD models with PSDs that account for volcanic aggregation in the eruptive column by using initial plume
models. FPLUME, a one dimensional (1D) plume model based on buoyant plume theory, constructs initial plume
characteristics that account for ash aggregation (Folch et al., 2016). In this case, the 1D plume model develops an initial PSD
at the source that accounts for aggregation processes and then keeps this PSD invariant during further plume transport.

In effort to study and predict volcanic ash aggregation effects using a fully coupled modeling system, where the fate
of the airborne ash particles is coupled to the atmospheric environment, a volcanic ash aggregation scheme was incorporated
into the Weather Research Forecasting with Chemistry (WRF-Chem) model (Grell et al., 2005). This coupled system
requires no temporal nor spatial interpolations as it calculates interactions between the meteorology and ash at each
modeling time step (on the order of seconds). While many dispersion models require less computing power than WRF, a
number of them require a mesoscale model, like WRF, to generate regional, gridded meteorological fields for their
initialization. As an example, FALL3D is typically initialized with a WRF model run that is executed prior to the dispersion
model. Modeling particle dispersion with WRF-Chem is, therefore, as computationally feasible as running these models
since in many cases, a mesoscale, gridded model must be run for their initialization.

The following sections of this paper detail the inclusion of a computationally feasible volcanic ash aggregation
scheme into the WRF-Chem model and the impacts of these modifications on model output. The following 'Aggregation
Parameterization and Implementation' section (Section 2) details the background and incorporation of a mathematical





scheme that is physically descriptive of aggregation processes into WRF-Chem, as well as the development of a new methodology for selecting aggregation sticking efficiencies that depend on relative humidity. This newly implemented code is then applied to the April and May 2010 eruptions of Eyjafjallajokull, as well as to a controlled sensitivity study using a single eruption. The setup of these two cases is discussed in Section 3 'Methods', with remarks on the model output in Section 4 'Results'. Concluding remarks and then provided in the final Section 5 'Conclusions'.

## 2. Aggregation Parameterization and Implementation

Smoluchowski (1917) developed the original analytical theory of the process of coagulation of colloid particles based upon Prof. R. Zsigmondy's experiments with gold solutions. The Smoluchowski Coagulation Equation (Eq. 1) is an integrodifferential, population balance equation that describes the evolution of particle number density, $n_v(v)$, in time $t$, as primary particles of one volume, $v$, collide and stick together with particles of different volumes, $v'$, to form aggregates (Smoluchowski, 1917). It is physically descriptive of the aggregation process.

$$\frac{\partial n_v(v)}{\partial t} = \frac{1}{2}\int_0^v K(v-v',v)n_v(v-v')n_v(v')dv' - n_v(v)\int_0^\infty K(v,v')n_v(v')dv' \tag{1}$$

Equation (1) describes the number of aggregates of volume $v$ formed, $n_v$, per unit time, $t$, on the left, and the loss of primary particles between volumes $v$ and $v'$ on the right as particles aggregate based on the collision frequency of the particles. Frequency is weighted by the coagulation kernel, $K$, which is the product of the collision kernel, $A$, and a sticking efficiency, $\alpha$, thus, $K = A\alpha$.

Volcanic ash may undergo various processes that result in collisions, such as Brownian motion, differential sedimentation and fluid shear, and as a result there are many formulations of the coagulation kernel, $K$ (Jacobson, 2005). For example, collisions due to Brownian interactions ($A_B$) occur randomly during diffusion and are temperature dependent. As temperature increases, the diffusion rate increases thus increasing their chances of interacting with other particles. Particle collisions due to shear ($A_S$) occur when ash moving in different horizontal directions collide due to changes in laminar flow. This kernel therefore depends on wind speed and direction. Lastly, differential sedimentation ($A_{DS}$) captures particle interactions due to the different fall velocities of different sized particles. The rate at which particles settle is dependent on their size and therefore the differential sedimentation kernel depends on the difference in size between particles. As larger particles fall, they have a greater chance of encountering smaller, slower moving particles on their descent. In summary, the collision kernels $A_B$, $A_S$ and $A_{DS}$ represent the rate at which ash particles collide based on Brownian motion, fluid shear and differential sedimentation, respectively. Each kernel depends directly on the number concentration and size distribution of ash particles, and each depends highly on its own set of parameters.

While physically descriptive of the aggregation process, the Smoluchowski Equation itself, in addition to the equations governing the coagulation kernel, $K$, is prohibitively computationally expensive to solve explicitly, even with simple boundary conditions. Advances in simplifying the equation for use in computational volcanic ash modeling resulted in large



part from work by Dekkers and Friedlander (2002) and Costa et al. (2010) by assuming a time independent aggregate size

distribution and fractal geometry of volcanic ash aggregates, respectively. Assuming a fractal aggregate geometry greatly

simplifies the equations describing the coagulation kernels ($A_B$, $A_S$ and $A_{DS}$) by establishing a particle size-volume fractal

relationship, described by a fractal dimension factor , $\xi$. In addition, an assumption of fractal geometry allows $n_v$ in Eq. (1) to

be described in terms of the total number of particles in a computational space, $n_{tot}$, forming aggregates of a certain fractal

dimension, $D_f$, based on a generally accepted fractal relationship (Jullien and Botet, 1987; Lee and Kramer, 2004). The

simplified Smoluchowski equation described by Costa et al. (2010) results in a calculation of $\frac{\partial n_v(v)}{\partial t}$, from Eq. (1), that is much

more computationally feasible (Eq. 2)

$$\Delta n_{tot} = \alpha \left( A_B n_{tot}^2 + A_s \Phi^{\frac{3}{D_f}} n_{tot}^{2-\frac{3}{D_f}} + A_{DS} \Phi^{\frac{4}{D_f}} n_{tot}^{2-\frac{4}{D_f}} \right) \Delta t \qquad (2)$$

Here, $\Delta n_{tot}$ represents the total number of particles per unit volume lost to aggregation. The equation relies on the

solid volume fraction of the aggregates, $\Phi$ (Folch et al., 2016), the number densities of the bins, $n_{tot}$, as well as the fractal

dimension of the fine ash particles, $D_f$ (Costa et al., 2010). Equations describing the collision kernel, $A$, were also simplified

using a fractal representation of ash geometry and were reduced to Eq. (3) through Eq. (5), shown in Table 1.

New code capable of calculating Eq. (2) to Eq. (5) was developed in this study and integrated into the Fortran 90

module "module_vash_settling.F" file, located in the "chem" subdirectory of the WRF main directory, which is available to

download from the WRF homepage: www2.mmm.ucar.edu/wrf/users/downloads.html. Modified code is available upon

request. See the following "Code Availability" section for details.

Most of the source variables necessary to solve Eq. (2) to Eq. (5) are available in WRF-Chem by selecting the

appropriate aerosol and chemistry packages. For example, chemistry option (chem_opt) 402 (WRF-Chem User Guide 3.9,

2018) includes chemistry and humidity variables provided by the Regional Deposition Acid Model Version 2 (RADM2)

(Stockwell et al., 1990) and the Goddard Chemistry Aerosol Radiation and Transport (GOCART) models (Chin et al., 2000),

as well as the inclusion of volcanic sulfur dioxide (SO₂) and ten volcanic ash particle size bins (Stuefer et al., 2013). Three

variables required by Eq. (2) to Eq. (5), the sticking efficiency, $\alpha$, fractal dimension, $D_f$, and fractal dimension factor, $\xi$ are

not, however, included in WRF-Chem and therefore must be calculated or assumed.

The fractal dimension, $D_f$, relates the number of primary particles $N$ in an aggregate to the size of the aggregate, $R$,

such that $N$ scales proportionally as $N \propto R^{D_f}$. For example, as $D_f$ approaches 3, primary particles in the aggregate use up

more and more space such that $D_f = 3.0$ would indicate a solid, filled aggregate. A lack of experimental data adds a degree

of uncertainty when selecting the fractal dimension, however previous analysis studies of aggregates selected after the

eruptive events from Mount Saint Helens and Mount Spurr suggested a dimension $D_f = 2.99$. This favorable fractal

dimension resulted from a regression analysis between model output and observed deposits (Folch et al., 2010). The fidelity





of confidence in the choice of the fractal dimension is hindered by the fact that it does not necessarily, by its definition,
remain constant within a plume.

         The fractal dimension factor, $\xi$, used to simplify the coagulation kernel equations relates the fractal dimension, $D_f$,
to the diameters and volumes of the primary particles in the aggregates. This relationship is given in Eq. (6)

$$d_i = \xi v_i^{\frac{1}{D_f}}$$
(3)

         Here, $d_i$ and $v_i$ are the diameter and volume of the primary particles forming an aggregate. Costa et al. (2010),
Dekkers and Friedlander, (2002) and Folch et al. (2010) indicate that a fractal dimension on the order of 0.6 to 1 is sufficient
for describing the geometry of volcanic ash particles and aggregates. As done in Costa et al. (2010), a unity fractal dimension
factor is utilized in this study.

         The sticking efficiency coefficient, α, relies heavily on the concentration of water vapor and ice (Costa et al., 2010).
In order to formulate an appropriate estimate for the sticking efficiency coefficient, a new parameterization was incorporated
into the WRF-Chem emissions driver that includes volcanic water vapor emissions that are specified by the user. This code
adds these emissions to the ambient water vapor mass within the model environment. Van Eaton et al. (2012) demonstrated
that the sticking efficiency of volcanic ash particles follow exponential curves. Using these fitted curves, the sticking efficiency
coefficient, α, between two particles $i$ and $j$ may be calculated using a fitting coefficient, $S$. This coefficient varies with water
vapor concentration, $[H_2O]$, and the radius of the colliding particles, $r$. A lookup table was added to select sticking coefficients
based on this work by utilizing the water vapor content of the model cell and the particle size (Eq. (7) and Table 2). Importantly,
this equation is computationally inexpensive to solve. Although electrostatic interactions are significant enough to cause
aggregation of particles, they are most likely insignificant when compared to aggregation in the presence of water (James et
al., 2003; Schumacher and Schmincke, 1995). Since the modeled background water rarely approaches 0% relative humidity,
dry interactions are not parameterized in this study.

$$\alpha(ij, [H_2O]) = e^{-Sr}$$
(3)

         The four aggregation equations (Eq. 2 to Eq. 5) are solved for volcanic ash bins 2 to 10 at every time step, for every
model grid cell, and account for interaction of particles between the different bins by using the total mass to calculate the
available number of primary particles available for aggregation. Large particles, greater than 1 mm in diameter, are included
in WRF-Chem volcanic ash bin 1, which has been designated as the "aggregate" bin. All aggregates generated by the code are
moved to bin 1 and their corresponding masses are subtracted from bins 2-10.  The large particles (in bin 1) assume high fall
velocities and contribute to ash fallout within periods of minutes (Rose and Durant, 2011). All volcanic ash removed from the
model domain is stored in the ASH_FALL variable, allowing the analysis of fallout mass and location.





## 3. Case Study and Methods

The 2010 eruption of Eyjafjallajökull Volcano in Iceland has been selected to test the modified WRF-Chem modeling experiment. Eyjafjallajökull erupted in April and May 2010, dispersing ash over Europe that caused numerous flight delays over the course of weeks and a resulting loss of revenue to airlines in the billions of dollars (Harris et al., 2010). Due to Eyjafjallajökull's location and the availability of observational resources, it became one of the most studied and well-documented eruptions in history, providing numerous sources of data regarding the plumes characteristics. The German

Aerospace Center (Deutsches Zentrum für Luft- und Raumfahrt, DLR) took several in situ measurements of Eyjafjallajökull's ash clouds over the course of the two months of eruptions by flying its Falcon aircraft into forecasted plume locations. Three of these flights are used for analysis in this study from 19 April, 16 May and 17 May. The flight paths corresponding to these flights are depicted using colored lines in Fig. 1. During the flights, Schumann et al. (2011) recorded particle number concentrations using a Grimm SKY-OPC 1.129 optical particle counter and a Particle Measuring Systems, Inc. (PMS) Forward

Scattering Spectrometer Probe (FSSP), observing a range of particles from 0.25 and 24 μm. In addition, upper and lower mass concentration estimates were calculated using the minimum and maximum imaginary component of the refractive index, of which the FSSP was particularly sensitive. For the flight of May 17, a medium estimate of mass concentration was calculated. From these studies, information on particle number, mass concentration, plume heights and gas composition are available, providing one of the best in situ datasets available to study distal and proximal volcanic emissions (Schumann et al., 2011). In

addition to these in situ data, Doppler measurements of the eruptive column and ground air sampling measurements were conducted by many groups to establish descriptive and accurate eruption source parameters (Arason et al., 2011; Devenish et al., 2012a, Devenish et al., 2012b; Stevenson et al., 2012). Observations of volcanic tephra fallout are also available and provide important insights into the PSD and transport of the distal Eyjafjallajökull ash clouds (Gudmundsson et al., 2012; Stevenson et al., 2012). In addition, volcanic ash aggregation was directly observed via high speed photography near the vent,

lending proof that particle aggregation occurred in the plumes Eyjafjallajökull produced (Taddeucci et al., 2011).

### 3.1. Eyjafjallajökull Model Domain Setup

       The newly implemented aggregation code was applied to the April and May 2010 eruptions of Eyjafjallajökull. Additionally, sensitivity studies were conducted using a hypothetic single eruption of Eyjafjallajökull on May 5[th], 2010. In all studies, the model domain was centered at 50°N, 0°W, offsetting the Eyjafjallajökull vent (63.62°N, 19.61°E) to the

northwest of the domain to account for the predominant southwest trajectory of the ash clouds. The model was setup for high spatial resolution simulations at 10 km$^2$ per grid cell, with a total of 500 x 500 horizontal grid cells. The domain is shown in Fig. 1 with Eyjafjallajökull marked in red. The model included 48 vertical pressure levels with the top level of the model set to 2,000 Pa. The integration time step of the dynamics and chemical fields was set to 30 seconds.

       Meteorological fields were obtained from the National Center for Environmental Prediction Final Global

Operational Analysis (NCEP FNL) datasets, ds083.2, accessed through the National Center for Atmospheric Research Data





Archive (NCAR, 2000). These datasets represent the final analysis of historical Global Forecast System (GFS) model
output. Ingest was conducted similar to Hirtl et al. (2019), using a 9 day spin up time before the first eruption on 14 April
and with meteorological initializations every 48 hours. The WRF-Chem volcanic package was enabled with chemistry option
402, which includes ten particle sizes of volcanic ash (Stuefer et al., 2013). These particle sizes are shown in Table 3. The
Yonsei University Planetary Boundary Layer (YSU PBL) scheme and the Noah Land Surface Model (LSM) were included
for PBL and near ground physics (Chen and Dudhia, 2001; Hong et al., 2006).

Water was added to the model domain by multiplying the water content of Eyjafjallajökull's magma, 1.8% (Keiding
and Sigmarsson, 2012) to the total erupted mass of 400 Tg for fine and coarse ash estimated by Taddeucci et al. (2011). This
1.8% multiplier produces water vapor emissions that agree with constraints constructed by comparing $H_2O/SO_2$ emission
ratios using values from Allard et al. (2011), yielding a ratio of 458 mol/mol, and $SO_2$ emission rates from two remote
sensing studies by Boichu et al (2013 and Thomas and Prata (2011). The code was modified to read in volcanic water vapor
emissions rates into WRF-Chem as a callable Fortran module.

In addition, Hirtl et al. (2019) noted that the model topography of Eyjafjallajökull is smoothed at the 10 $km^2$ model
spatial resolution, resulting in a vent height 400 m lower than the actual height of 1000 m. A 400-m height offset was applied
to correct this.

**3.2. Sensitivity Study Model Setup**

Multiple sensitivity studies were conducted in order to assess: 1) the overall change in mass due to aggregation, 2)
the effects of different fractal dimensions, $D_f$, on the aggregation rate, 3) the contribution of each collision kernel, $A_B, A_S$ and
$A_{DS}$, to the decrease in domain ash mass and 4) the effect of adding coupled water vapor emissions to the model domain on
the aggregation rate. These sensitivity studies were conducted on a smaller time slice of the parent domain, using a 9 hour
eruptive event on May 5th, 2019, initialized at 00:00Z with a rate of 4 x $10^6$ kg $s^{-1}$, which corresponds to an average value of
Eyjafjallajökull's largest eruptions. A 72 hour spin up time was included prior to the eruption initialization to allow the
meteorological fields to stabilize, and was then run for 6 days, ending 00:00Z on the 11th of May. New meteorological fields
were ingested every 24 hours for high fidelity. Each volcanic ash bin was populated with 10% of the total erupted mass in
order to simplify output analysis.

In order to assess how the aggregation code affects model output, WRF-Chem was run with and without the
aggregation code enabled. Due to a lack of experimental data, a choice of fractal dimension, $D_f$, is difficult. Therefore, the
fractal dimension, $D_f$, was varied to measure its effects on the overall aggregation rate. The span of fractal dimensions
chosen ranges from $D_f$ = {2.5, 2.6, 2.7, 2.8, 2.9, 2.95, 2.98, 2.99, 3.0} and is based on studies by Costa et al. (2010) and
from a similar study of Mount Saint Helens and Mount Spurr using Fall3D by Folch et al. (2010).

The contribution of each collision kernel, $A_B, A_S$ and $A_{DS}$, to the total reduction in domain mass was also assessed
by using the same domain and eruption parameters, and enabling only one kernel at a time using a fractal dimension of 2.5





and 3.0. The total change in mass from each kernel was then divided by the total change in mass with all kernels enabled to find the percent contribution.

255       The impacts of the inclusion of water vapor on the aggregation rate were studied by running the code with and without the 1.8% water vapor emissions included in the model domain. For the simulation run without water vapor emissions, only background water vapor from the FNL datasets were used.

### 3.3 Model Setup for April and May 2010 Eruptions of Eyjafjallajökull

WRF-Chem was also configured to simulate Phase I (April 14-18, 2010) and the Phase III (May 4-18 2010)
eruptions of Eyjafjallajökull using the same model domain described above. Phase II eruptions were effusive rather than explosive and ejected tephra at much lower altitudes of 2 to 4 km ASL (Gudmundsson et al., 2012) and were thus not included in this modeling case study.

Eruption source parameters (ESP) for Eyjafjallajökull were adapted from Mastin et al. (2014) and Hirtl et a. (2019). Camera footage and C-band Doppler radar measurements were used to establish three hourly plume heights for the April and
May 2010 eruptions (Arason et al., 2011; Mastin et al., 2009; Hirtl et al., 2019). These plume heights were used to calculate eruption rates based on the plume height/eruption rate relationship derived by Mastin et al. (2009). The total erupted mass was then scaled based on work by Gudmundsson et al. (2012) such that the total ash mass ejected over the eruptive phases agreed with the 170 Tg Phase I estimate and 190 Tg Phase III estimates for fine ash stated (Hirtl et al., 2019). The bimodal, silicic (S2) ESP particle size distribution (Table 3.3) was used to populate the ten volcanic ash bins in the model (Mastin et
al., 2009). The three hourly plume heights and eruption rates used in the study are presented in Fig. 2.

In this study, all aggregation collision kernels were enabled, and water vapor emissions as described previously were added to the model domain at each time step. As mentioned earlier, the choice of a fractal dimension is hindered by a lack of experimental data. Folch et al. (2010) conducted linear regression analysis of repeated model run comparisons to tephra fallout measurements from eruptions originating at Mount Spurr and Mount Saint Helens. This study resulted in the
use of a $D_f = 2.99$ fractal dimension. Due to a lack of experimental data on the development of volcanic ash fractal dimensions, and the fact that aggregate fractal dimensions are not necessarily constant with time, $D_f$ was set at the upper bound of 3.0, providing a maximum effect of particle aggregation.

### 4. Results

The newly implemented aggregation parameterization was first assessed with a sensitivity study of a singular eruptive
event, and then by application to the entire Phase I and Phase III eruption periods.



## 4.1. Sensitivity Study Results

Varying the fractal dimension between 2.5 and 3.0 resulted in a range of aggregation rates. Figure 3 illustrates the change in domain mass from a single 9-hour eruption on May 5th at 00:00Z with a constant eruption rate of 4 x 10^6 kg s^-1. As expected, higher values of $D_f$ result in higher rates of aggregation with the largest jumps in the aggregation rate between $D_f$ = 3.0 and 2.8. The degree to which aggregation reduced the overall ash domain mass can be seen in the peak mass loadings at hour 9 in Fig. 3. Here, the peak domain mass using $D_f$ = 3.0 is 17.4 Tg. This is 72% reduction in peak mass compared to the non-aggregation enabled run of 62.9 Tg. Lower values of $D_f$ provide almost no change in the total domain mass. For example, $D_f$ = 2.5 results in a 0.7 % decrease in peak mass by about 0.5 Tg.

To quantify the change in aggregation rate, volcanic ash lifetimes in terms of e-folding were calculated. This analysis is presented in Fig. 4 and indicates a range of e-folding times from 72 hours with no aggregation code enabled to 15 hours with maximum aggregation considered ($D_f$ = 3.0). As the fractal dimension increases, the atmospheric lifetime of volcanic ash decreases due to the incorporation of more volcanic ash particles into each aggregate. When considering fractal dimensions 2.7 and lower, the total lifetime is reduced only slightly, less than 4%. Larger decreases in lifetime become apparent with $D_f$ = 2.8 (10% decrease) and jump thereafter to a maximum 79.5% decrease at $D_f$ = 2.99 and $D_f$ = 3.0 (same decrease for both). Based on work by Folch et al. (2019), it is assumed that an optimal value of the fractal dimension likely lies near $D_f$ = 2.99, which corresponds to a 79.5% difference in e-folding times. In terms of volcanic ash lifetime, on hourly timescales, there is no difference between $D_f$ = 3.0 and 2.99.

Figure 5 shows the extent to which each kernel contributed to the overall change in the model domain's ash mass by enabling each kernel independently. Two fractal dimensions were considered, $D_f$ = 2.5 and 3.0, and both affected each kernel's contribution to aggregation differently. The differential sedimentation kernel, $A_{DS}$, for example contributed to the majority of the change in domain mass over the course of the 96-hour model run ($\approx$ 99%) when $D_f$ was set to 3.0, but contributed only 5% on average with $D_f$ = 2.5. The Brownian kernel became the major contributor to aggregation in the case of $D_f$ = 2.5, contributing to over 90% of the aggregation. This agrees with parametric studies of varying fractal dimensions by Costa et al. (2010), who noted this trade between $A_{DS}$ and $A_B$ when considering fine ash particles (<63 $\mu$m). Overall, fluid shear interactions were the minor contributor to aggregation for both fractal dimensions. While its contribution to aggregation approaches that of $A_{DS}$ for $D_f$ = 2.5, it is many orders of magnitude lower than $A_B$ or $A_{DS}$ for $D_f$ = 3.0.

Figure 6 illustrates the total domain mass for fine ash (bins 7-10) in panel (a) as well as their percentage of total domain mass in panel (b), representing the PSD of the fine ash fraction. Figure 6 considers maximum aggregation with $D_f$ = 3.0. The bins with larger ash particles (1-6) were not included due to the rapid decrease in their domain mass as a result of their high settling velocities. Figure 6 (a) depicts a decreased mass loading for each bin when aggregation is enabled, as well as a shorter lifetime, as expected. Figure 6 (b) depicts a shift in the particle size distribution due to aggregation. The



aggregation code results in less contribution from fine ash particles (bins 7, 8 and 9), resulting in a shift of the PSD towards bin 10. Bin 10 in the aggregation enabled code makes up an extra 10% of the model domain mass upon reaching near steady

state at model hour 120. This is the result of the increased aggregation of the larger sizes particles since larger radii result in a larger probability cross section of collision and subsequent aggregate formation.

Coupling water emissions resulted in a very small increase in aggregation rate, lowering the total domain mass on the order of Mg hr⁻¹, much lower than the overall loss rate of ash due to aggregation on the order of Tg hr⁻¹ (6 orders of magnitude). The sticking efficiency, Eq. (6), is high (> 90%) for small particles (< 63 $\mu$m). As the residence time of large particles is very

short, the sticking efficiency is applicable to the narrow range of particle sizes that persist in the domain (Bins 7-10, < 32.5 $\mu$m). These particle sizes correspond to a narrow range of sticking efficiencies (.87 to .97), regardless of the water vapor concentration.

### 4.2. Eyjafjallajökull Study Results

The ash cloud dynamics generated by WRF-Chem over the model period agree with other modeling studies of

Eyjafjallajökull utilizing WRF-Chem (Hirtl et al., 2019; Webley et al., 2012). Figure 7 provides an example of the output from WRF-Chem for April 15 and 16, 2010. The dynamics of the ash clouds are apparent. The plume moves south and east towards the coasts of Scandinavia and northern Europe then splits into two plumes: one residing over Sweden and Finland and the other passing through multiple northern European countries.

Model output also agrees with airborne in situ measurements. The DLR research aircraft conducted 13 flights on 11

different days that transected Eyjafjallajökull's ash clouds over the course of the Phase I and Phase III eruptions (Schumann et al., 2011). Predicted ash concentrations from WRF-Chem were compared to the in situ observational data from three of these flights: April 19 and May 16, and 17, 2010. WRF-Chem volcanic ash bins 8, 9 and 10 correspond to the particle size detection limits of the Grimm OPC and PMS FSSP aboard the Falcon aircraft and were thus chosen for comparisons.

Figure 8 presents time series plots of WRF-Chem output and DLR measurements. Figures 8(a), 3.9(c), and 8(e) cast

the WRF-Chem output in mass concentration (g m⁻³). Figures 8(b), 8(d) and 8(f) cast the WRF-Chem ash bin as number concentrations by using an assumed particle density of 2500 kg m⁻³ (Brown et al., 2012) in order to make direct comparisons to the Grimm OPC and FSSP detectors.

Temporal changes in observed and modelled ash concentrations agreed moderately well for the April 19 flight (Fig. 8a and 8b). Analysis of particle number densities in Fig. 8 (b) for April 14 shows 5 significant overestimations of volcanic

ash by the non-aggregation enabled code, between 50-75% at 14:55 and 15:07, between 15:15-15:18, between 15:35-15:42 and between 16:55-17:06. These overestimations did not occur when the aggregation code was used. One peak concentration was observed at 15:30 UTC on April 19, which was not resolved by WRF-Chem (Fig. 8b). Typical of any Eulerian air quality model, WRF-Chem tends to diffuse ash concentrations, an effect that is also dependent on the model resolution.



Smaller domain grid cells permit better comparison with point observations, but decreases in grid cell sizes are

computationally expensive.

Number density readings for May 15 (Fig. 8d) contained more robust data than mass concentration (9C) and was therefore used in the analysis. Here, a large overestimation of ash is calculated by WRF-Chem when not using the aggregation code. A peak of 290 particles cm$^{-3}$ are observed in the unmodified code, almost 10 times higher than observed. With aggregation enabled, the WRF-Chem solution is much closer to the observed numbers at a maximum of 45 particles cm$^{-3}$.

On May 17 (Fig. 8e and 8f), the aircraft performed a steep transect through a plume with larger ash particles. Almost no ash concentration was recorded at the lowest flight altitude reached during the middle of the flight at 16:40 UTC. At this same time, WRF-Chem predicted concentrations in excess of 400 g m$^{-3}$. Where the plume locations do agree, there is improved agreement between the aggregation enabled code and the airborne observations of mass concentration. For the entire time range, observations where the aggregation code produced mass readings in the same order of magnitude as those

observed by DLR were counted. This total was then divided by the total flight time and resulted in an average 80% agreement of the data (78% for April 19, 78% for May 15 and 83% for May 17). This fell to an average of 62% when the code was run without aggregation, using the same methodology.

In addition to comparisons with Schumann et al. (2011) in situ measurements, WRF-Chem tephra fallout was also compared to field measurements of tephra collected by Stevenson et al., (2012) in the United Kingdom (UK). Figure 9

depicts the mass of tephra deposited in the model domain from all April 2010 eruptions in panel (a) and from May 2010 eruptions in panel (b). Stevenson et al. (2012) report three sampling periods that overlap with the model domain times in this study. For example, Stevenson et al. (2012) counted 218 grains of tephra per cm$^2$, at Benbecula in the Outer Hebrides (57.43N, 7.34W, Fig. 9(a), white circle), with a mean diameter of 18 ± 7 $\mu$m while sampling between 13-20 May, 2010. Assuming an average density of 2,500 kg m$^{-3}$ yields a tephra concentration between 20 and 45 mg m$^{-2}$, compared to 31 mg

m$^{-2}$ predicted by WRF-Chem with the aggregation code enabled during the same time range. Samples taken at Leicestershire (52.73°N, 1.16°W, Fig. 9(b), white circle) between 25 April and 3 May, 2010 estimate a range of tephra mass on the ground between 51 and 119 mg m$^{-2}$, also near the WRF-Chem estimate of 41 mg m$^{-2}$ (80% of observed mass) between those dates. Another sample from Lincolnshire (52.74N, 0.38W, Fig. 9(b), white circle) covered a period from 24-30 April 2010. In this case, tephra fallout between 3 and 13 mg m$^{-2}$ were measured, whereas WRF-Chem predicted a smaller value of 1.2 mg m$^{-2}$

(40% of observed mass). The smaller estimates for the Lincolnshire and Leicestershire sites may be explained by the lack of model data covering April 27 –May 3, as the last modeled hour was 00:00 UTC on April 27. When considering WRF-Chem run without aggregation, the modeled fallout seen in these areas is minimal, with less than 1 mg m$^{-2}$ observed.

The aggregation code altered the total domain mass of each volcanic ash bin. To study this change, the model domain mass was analyzed from May 14 to 18, 2010. This time frame represents the last 96 hours of modeled eruptions and

includes a high degree of variability in the eruption rate and plume height (see Fig. 2). The total domain mass is presented in Fig. 10 without (a) and with (c) the aggregation code enabled. To analyze the PSD, the mass of each volcanic ash bin was divided by the total model domain mass. The resulting percentages are presented in Fig. 10(b) and 11(d). The top panels,



Fig. 10(a) and 10(b), depict WRF-Chem output without the use of the aggregation code, whereas the lower panels, Fig. 10(c) and 10(d), include the aggregation code. The short atmospheric lifetime of the large particles in bins 1-3 result in small

masses during this time frame compared to bins 4-10 including smaller particle sizes. As such, only bins 4-10 are depicted in Fig. 10. Major changes in the eruption rates are annotated on the time axis with red marks.

Two important observations are noted when aggregation is included. First, the total domain mass in each bin is reduced and second, the PSD shifts towards smaller sized particles during eruptive events. For example, the initial period in Fig. 10 is eruptive until the first red mark on the 14th at 09:00UTC. During this period, the eruption rate is 7.36 x$10^5$ kg s$^{-1}$

(7.949 Tg per 3 hours). In the non-aggregation enabled code, the dominant ash species are bins 6, 7 and 8 which have peak masses of 3.7, 4.1 and 3.3 Tg, respectively. In the non-aggregation enabled code, bins 6, 7 and 8 make up the majority of the domain as mass, contributing 21.5%, 24.1% and 19.3% of the total domain mass. When the aggregation code is enabled, the total domain mass for each of the bins is reduced to 1.0, 1.5 and 1.4 Tg, respectively, which is around one third of the original peak mass, showing an overall reduction. Additionally, their contribution to the overall domain mass changes to

14.6%, 21.1% and 20.5%. The smaller bin 8 ends up with more of the mass, with the other two contributing less to the PSD. In fact, the smaller bins 9 and 10 also contribute more to the overall domain mass, increasing from a peak of 13.1% and 9.6% on the 14th and 09:00UTC without the aggregation code enabled to 15.2% and 11.6% with the aggregation code enabled. Overall there is a slight shift towards smaller particle bins during eruptive events.

Interestingly, this trend in the PSD is not observed during periods of decreased eruption rates, while trends in overall

domain mass continue are still observed. Between marks 1 and 2, the eruption rate decreases from 7.36 x $10^5$ kg s$^{-1}$ to 1.09 x $10^5$ kg s$^{-1}$. During this period of slower eruption rates, the total domain mass continues to increase, however it is much lower when aggregation is considered. The PSD, on the other hand, remains consistent, with bins 8, 9 and 10 trending similarly in the non-aggregation and aggregation enabled case. This suggests that the aggregation code is most effective during eruptive events when particles are in high concentration.

Without aggregation, the only sinks for volcanic ash are via settling or via the plume traveling out of the model domain. For finer ash particles, removal via settling is minimal when compared to larger particles which is evident in Fig. 10(a) and 10(c). During periods of less volatile eruptions, such as between markers 1 and 2 or markers 3 and 4, the fine ash bins reach a steady state where the source of ash is almost equal to the sink, i.e. settling. This is evident in the horizontal slope of the bin domain mass. This is not true for larger particles whose settling velocities are high enough to remove them

faster than they are added. Aggregation adds an additional sink that is noticed subtly during less eruptive phases as the slight dips in domain mass, as well as the more pronounced decreases in the slope of the change in domain mass during periods of higher eruption rates.



## 6. Summary and Conclusions

A parameterization of volcanic ash particle aggregation has been implemented into the fully coupled WRF-Chem model. The new model has been tested for ash loadings and lifetimes. A simplified version of the Smoluchowski coagulation equation (Costa et al., 2010; Dekkers and Friedlander, 2002; Folch et al., 2010, 2016; Smoluchowski, 1917) was incorporated into the WRF-Chem model. This simplified method was chosen for its computational efficiency, allowing the aggregation rate to be calculated at each model time step in line with the atmospheric dynamics.

The effects of the aggregation code were assessed by applying it to a high-resolution model study of the 2010 eruptions of Eyjafjallajökull, including a single study of a 9 hour test eruption. The effect of each particle collision kernel on the overall aggregation rate (Eq. 2) was studied. The degree to which each kernel affected aggregation depended on the choice of the fractal dimension, $D_f$. The differential sedimentation kernel provided the largest contribution by orders of magnitude when a fractal dimension of 3.0 was chosen, however the Brownian kernel dominated when a fractal dimension of 2.5 was chosen. This result suggests that vertical motion, when a fractal dimension near 3.0 is chosen, is the primary driving force behind particle interactions in the aggregation process, rather than random (Brownian) or horizontal (shear) motions. Additionally, analysis of the volcanic ash lifetime shows that varying the fractal dimension may greatly vary the lifetime, especially when considering fractal dimensions between 3.0 and 2.8.

The Eyjafjallajökull model study was assessed by comparison to aircraft in situ measurements taken by DLR as well as tephra fallout samples measured in the United Kingdom. By comparing WRF-Chem calculated volcanic ash mass concentrations using the aggregation code to those observed by DLR, an average 80% match in an order of magnitude was observed for the 3 flights analyzed. Additionally, non-aggregation enabled code calculated 20-50% higher volcanic ash concentrations on numerous occasions, where the aggregation enabled code did not. The aggregation enabled WRF-Chem code tended not to overestimate volcanic ash, or to overestimate less than the non-aggregation enabled version, potentially yielding more realistic ash concentrations which may benefit aircraft hazard mitigation forecasting.

As the plume transported over the United Kingdom, WRF-Chem predicted ash fallout that compared well to field measurements . Tephra fallout generated by WRF-Chem fell within observed values at one sample location, and predicted on average 60% of the fallout at two others. This suggests that WRF-Chem may be used to model not only the atmospheric transport of ash clouds, but the deposition of ash as well.

Importantly, these observations all suggest that two factors drive volcanic ash aggregation when including aggregation in the WRF-Chem code. First, volcanic ash concentration is noted to be the primary driving factor behind aggregation rate. The majority of model domain mass decreased near the vent where concentrations of ash are high. In addition, PSD analysis indicates that bins with higher portions of the eruption PSD undergo faster rates of ash aggregation. Bins with a larger share of the eruption PSD will aggregate faster due to their increased probability of collision. Second, vertical motions of ash falling through the atmosphere also drive the aggregation process through differential sedimentation for realistic ranges of fractal dimension (between 2.95 and 3.0).



The inclusion of this aggregation scheme into WRF-Chem provides research and operational meteorological communities a second VATD model to Fall3D that includes volcanic ash aggregation and is the first to run aggregation in an inline fashion where aggregation equations are solved at each model time step (Folch et al., 2010). This inline computation of volcanic ash yields many benefits. For example, the code identifies the driving forces behind volcanic ash aggregation, i.e. ash

concentration and differential sedimentation rates, and allows for the study of the effects of water vapor concentration on the aggregation rate. In addition, it allows the study of changes in particle size distributions due to enhanced ash settling as a result of aggregation processes, which are of particular importance to remote sensing communities where the effective particle size directly impacts the spectral methods used for detection. The modified code also benefits the operational volcanic ash modeling community by providing another VATD model for use in aircraft hazard mitigation. Additionally, the modified code is

computationally expedient. It ingests global models and runs volcanic ash dispersion and aggregation code while simultaneously calculating mesoscale atmospheric dynamics, eliminating the need for additional, offline dispersion runs. Ultimately, this study provides another step towards the inclusion of volcanic ash aggregation, an important physical process, into VATD models.

**7. Acknowledgements**

This publication is the result of research sponsored in part by the NOAA Cooperative Institute for Alaska Research (CIFAR) with funds from NOAA under cooperative agreement NA13OAR4320056 with the University of Alaska Fairbanks (UAF). The Alaska Space Grant Program supported this work. Computational time was provided by UAF Research Computing Systems at the Geophysical Institute and the Department of Defense High Performance Computing and Modernization Program.






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




**Figures**

**Figure 1**: WRF-Chem model domain used for simulations in Lambert Conformal projection with true latitude and longitude and
center at 0°E/W, 50°N. Location of Eyjafjallajökull (63.62°N, 19.61°W) marked with red dot. DLR Falcon flight paths for flights
on April 19 (red), May 16(green) and May 17(blue) shown with colored lines.


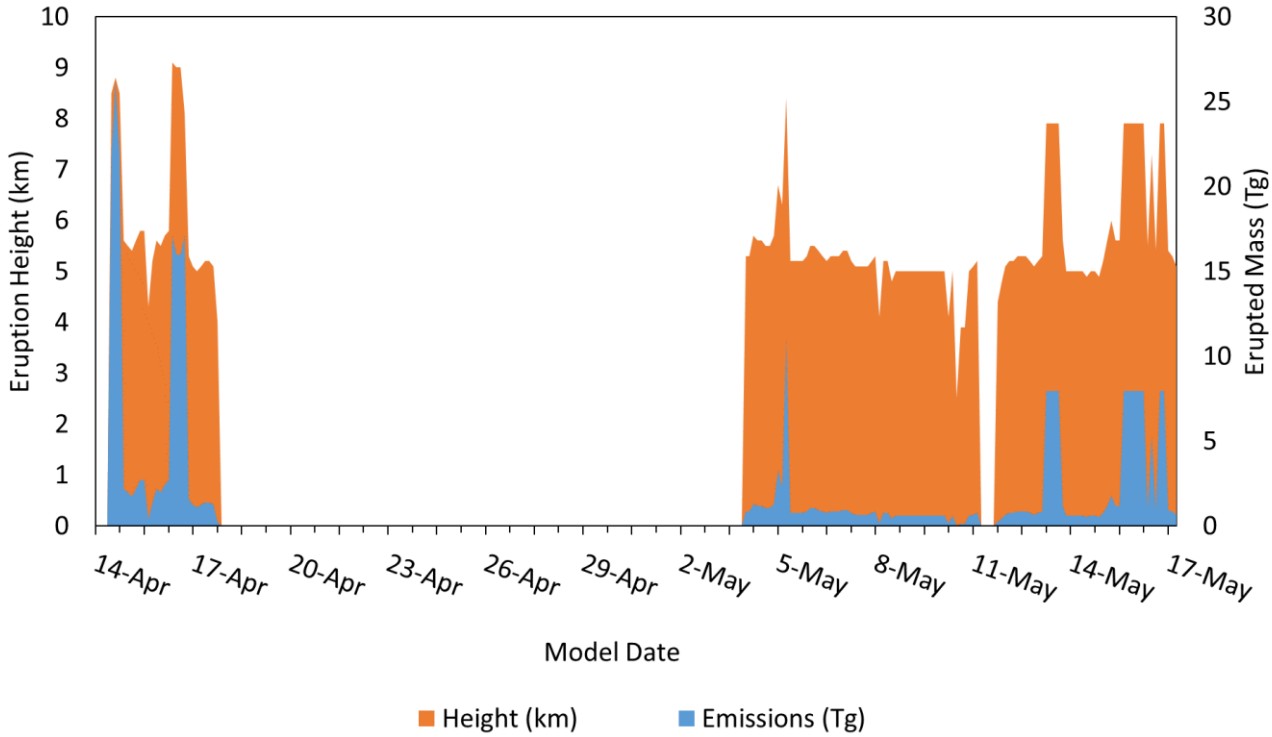

**Figure 2**: **Three hourly plume heights (KM) ASL (orange, km) and emitted mass (blue, Tg) used in the WRF-Chem modeling simulations (volc_d01.asc name list) for the eruption period April 12 until May 18, 2010. Values adapted from Hirtl et al., (2019) with dates as DD/MMM.**



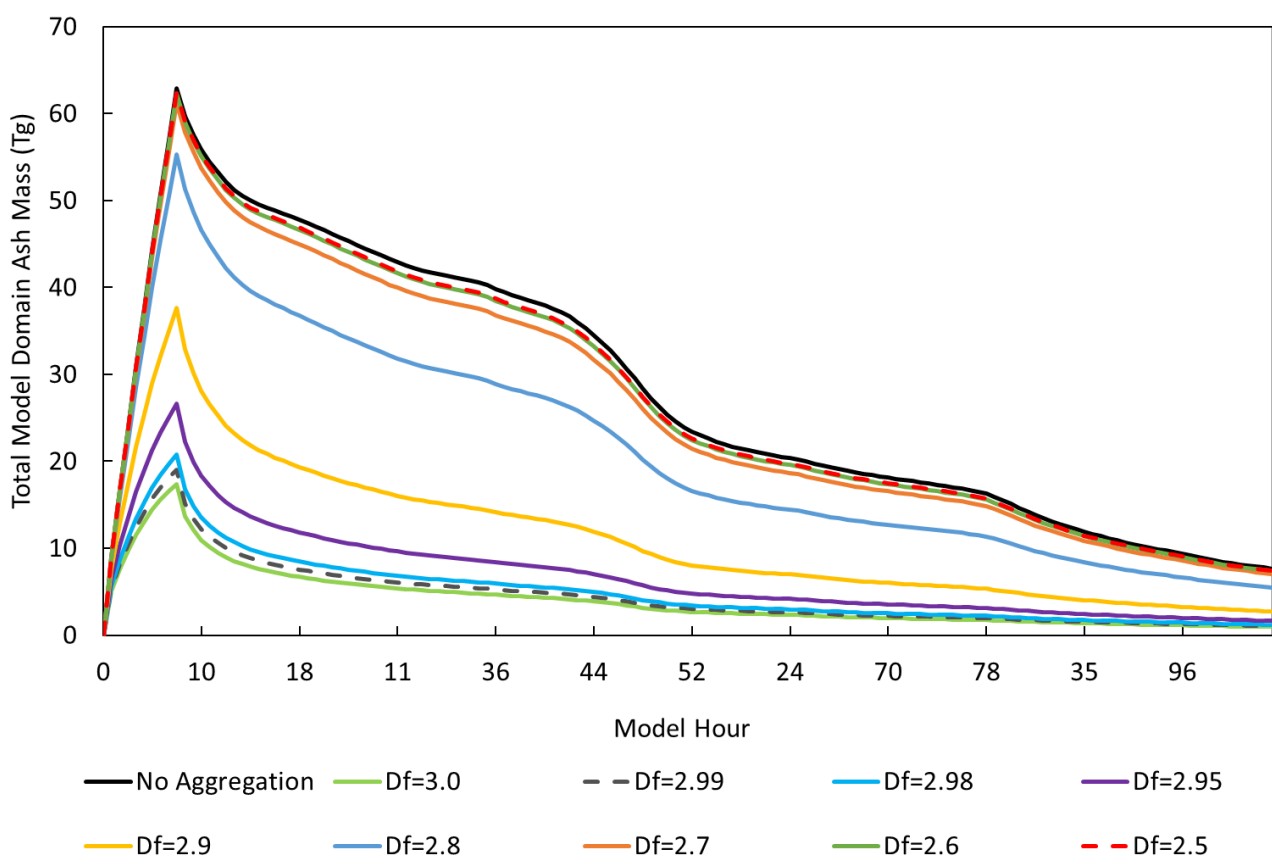

**Figure 3:** Change in total domain ash mass, Tg, for a hypothetical eruption on May 5th, beginning 00:00Z and ending 09:00Z, for a range of fractal dimensions, $D_f$ = {3.0, 2.99, 2.98, 2.95, 2.9, 2.8, 2.7, 2.6, 2.5}. Constant eruption rate = $4 \times 10^6$ kg s$^{-1}$.

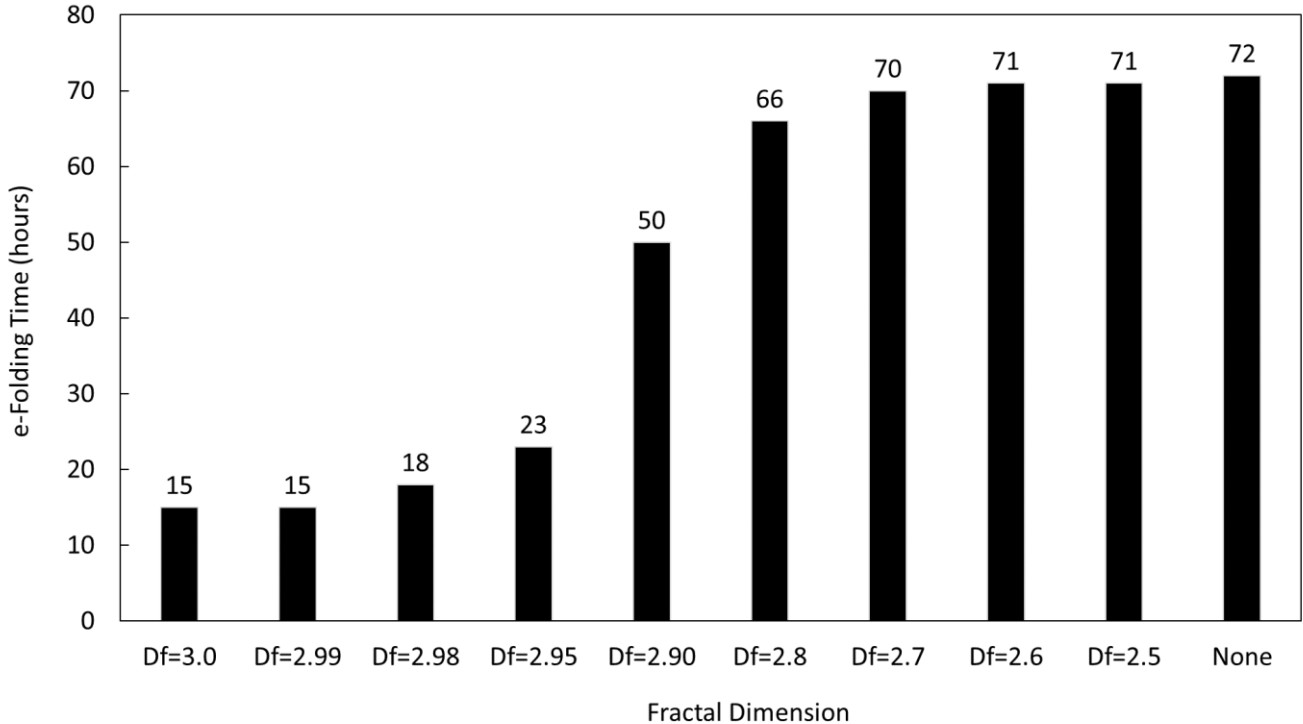

**Figure 4:** Volcanic ash e-folding time in hours for a hypothetical eruption on May 5[th], beginning 00:00Z and ending 09:00Z, for a range of fractal dimensions, $D_f$ = {3.0, 2.99, 2.98, 2.95, 2.9, 2.8, 2.7, 2.6, 2.5}. Constant eruption rate = 4 x 10[6] kg s[-1].



**Figure 5**: Percentage of aggregation rate for each collision kernel ($A_B$ = Brownian, $A_S$ = Shear, $A_{DS}$ = Differential Sedimentation) when considering a hypothetical eruption on May 5th, beginning 00:00Z and ending 09:00Z, for two fractal dimensions, $D_f$ = {3.0, 2.5}. Constant eruption rate = 4 x $10^6$ kg s$^{-1}$.




**Figure 6:** Total domain mass (A) and particle size distribution (B) of volcanic ash bins 7 to 10 when considering a hypothetical eruption on May 5th, beginning 00:00Z and ending 09:00Z, and a fractal dimensions, $D_f$ = 3.0. Constant eruption rate = 4 x 10$^6$ kg s$^{-1}$.



**Figure 7**: WRF-Chem generated volcanic ash column densities for the Eyjafjallajökull eruption in April 2010 at four hour intervals,
A = April 15 at 08 UTC, B = April 15 at 12 UTC, C = April 15 at 16 UTC, D = April 15 at 20 UTC, E = April 16 at 00 UTC, F = April
16 at 04 UTC, G = April 16 at 08 UTC, H = April 16 at 12 UTC, and I = April 16 at 16 UTC. Note each time output is at 00 hr.


**Figure 8: Comparisons of WRF-Chem model output to in situ mass concentrations (left panels) and particle numbers (right panels) observed by DLR during April 19 (A and B), May 15 (C and D) and May 17 (E and F), 2010 flights.**





**Figure 9** **Mass of tephra fallout deposited on model surface, lowest model level in WRF-Chem, for April (A,) and May (B) 2010 model simulations. White circle in (A) marks the Outer Hebrides and white circle in (B) marks Lincolnshire and Leicestershire, UK, corresponding to sample areas in Stevenson et al., (2012). Maximum domain fallout is 52 Mg m$^{-2}$.**


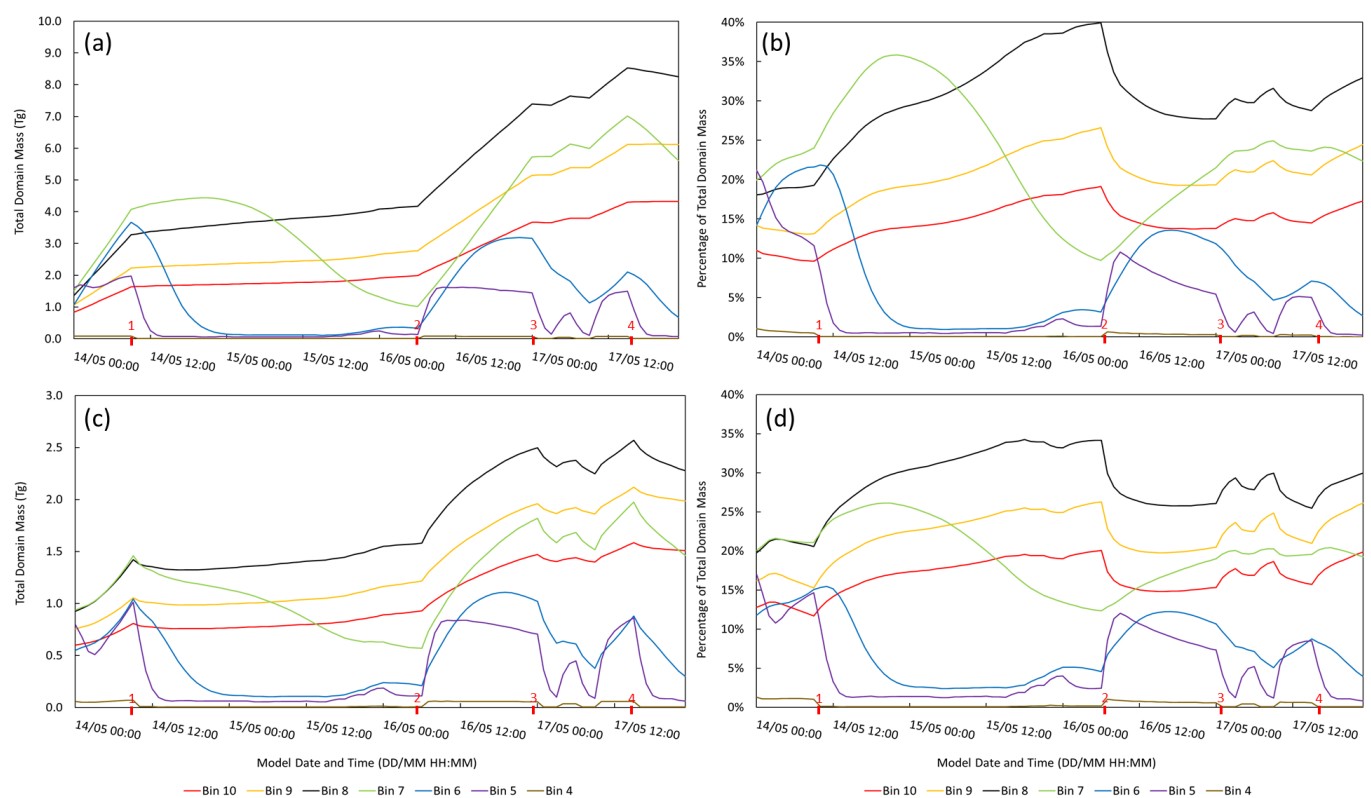

**Figure 10**: Total domain ash mass (A, C) and percent contribution to domain mass (B, D) for the modeled period between 14 and 18 May, 2010 without (A, B – upper panels) and with (C, D – lower panels) aggregation code enabled. Red numbers on date/time axis denote major (> 10%) changes in the eruption rate: 1) 14/09Z - Decrease from 7.949 to 1.775 Tg/3 hours, 2) 16/06Z - Increase from 1.175 to 7.949 Tg/3 hours, 3) 17/00Z - Decrease from 7.949 to 1.056 Tg/3hours, 4) 17/15Z - Decrease from 7.949 to 0.966 Tg/3hours. Note there are variable increases and decreases in the eruption rate between times 3 and 4.




**Tables**

**Table 1** – *Derived coagulation kernel equations used in the calculation of ΔN.*

| Kernel | Equation | (#) | Variables and Units |
|---|---|---|---|
| **Brownian Motion** | $A_B = \dfrac{4kT}{3\mu}$ | (3) | $k_b$ – Boltzmann Constant - m$^2$ kg s$^{-1}$ K$^{-1}$ <br> T – Temperature – K <br> $\mu$ – Dynamic Viscocity - kg m$^{-1}$ s$^{-1}$ <br> $d$ – Diameter - m |
| **Fluid Shear** | $A_S = -\dfrac{2}{3}\xi^3 \varGamma_S$ | (4) | $\varGamma_s$ - Fluid Shear – s$^{-1}$ <br> $d$ – Diameter – m <br> $\xi$ – Fractal Dimension Factor |
| **Differential Sedimentation** | $A_{DS} = \dfrac{\pi(\rho_p - \rho)g}{48\mu}\xi^4$ | (5) | $d$ – Diameter – m <br> $\xi$ – Fractal Dimension Factor <br> $\rho$ – Density of Air <br> $\rho_p$ – Density of primary particle <br> $V_d$ - Fall Velocity – m s$^{-2}$ |




**Table 2** – *Ash aggregation coefficients based on liquid water content, w/w, as described in Van Eaton et al., (2012). The weight percent of water (w/w) is calculated as mass of water divided by mass of the atmosphere.*

| Liquid Water Content (w/w) | Corresponding S value |
|---|---|
| 0% (ice) | 0.020 |
| 0-10% | 0.008 |
| 10-15% | 0.004 |
| 15-25% | 0.002 |






**Table 3** – *Distribution of volcanic ash in model domain among 10 size bins corresponding to the S2 size distribution as given in Mastin et al. (2009). The percentages of mass per bin are specified in the volc_d01.asc name list and may be given any value between 0 and 100. .*

| Bin | Diameter | Percent Mass |
|-----|----------|--------------|
| 1 | 1-2 mm | 22.0 |
| 2 | 0.5-1 mm | 5.0 |
| 3 | 0.25-0.5 mm | 4.0 |
| 4 | 125-250 μm | 5.0 |
| 5 | 62.5-125 μm | 24.5 |
| 6 | 31.25-62.5 μm | 12.0 |
| 7 | 15.625-31.25 μm | 11.0 |
| 8 | 7.8125-15.625 μm | 8.0 |
| 9 | 3.9065-7.8125 μm | 5.0 |
| 10 | <3.9065 μm | 3.5 |




**Code Availability**

This work modified the Weather Research Forecasting with Chemistry (WRF-Chem) base code. This included the creation and modification of text and Fortran files that replace and augment existing WRF-Chem code. These files may be accessed using the DOI reference provided upon publication of the code at doi:10.5281/zenodo.3540446. Code modifications along with descriptions are also directly available from the author upon request.
