# Peer review of "Modeling volcanic ash aggregation processes and related impacts on the April/May 2010 eruptions of Eyjafjallajökull Volcano with WRF-Chem."

_Natural Hazards and Earth System Sciences, 2019_

## Referee Comment (RC1) · Anonymous Referee #1 · 23 Dec 2019

This work covers the very important topic of volcanic ash aggregation that must be taken into account when modeling the dispersion of volcanic clouds. Aggregation of volcanic particles shapes the size distribution of particles in the traveling plumes and affects the long range transport of ash with important implications for aviation safety. The authors develop a physically based WRF module to describe such processes with improved results compared to observations. Overall the paper is important and well written and I suggest publication with a few minor comments as seen below:

Specific comments

– Is the 10x10 km grid size adequate to discribe the near-source aggregates (<15km

distance from the volcano)? Could you consider a nested higher resolution grid over Iceland to address these processes? This may not be important for the long range transport since anyhow the bigger particles will be removed from the model but it could provide more insight on the processes and probably improved deposition fluxes near the erruption.

– Please check that the references are provided in chronological order throughout the text.

– In line 92 " As an example, FALL3D is typically initialized with a WRF model run that is executed prior to the dispersion model. Modeling particle dispersion with WRF-Chem is, therefore, as computationally feasible as running these models since in many cases, a mesoscale, gridded model must be run for their initialization". Indeed, but you can run multiple faster Lagrangian dispersion simulations with different configurations using a single meteorological output (e.g. WRF) which may be important for determining aviation hazard under different emission scenarios.

– "One peak concentration was observed at 15:30 UTC on April 19, which was not re-solved by WRF-Chem (Fig. 8b). Typical of any Eulerian air quality model, WRF-Chem tends to diffuse ash concentrations, an effect that is also dependent on the model res-olution." I suggest that you should elaborate more on this mismatch between model and observed ash concenntrations. Such high peaks are the primary threat for aviation and moreover these are observed at about 2km elevation which may imply approach or takeoff heights thus increasing the potential danger. This may not be due to Eulerian diffusion otherwise one would expect a more uniform reduction of the concentration fields. Could you please check the concentration at the surounding gridpoints to check if possibly such concentrations exist and are misplaced by the model ?

– "Without aggregation, the only sinks for volcanic ash are via settling or via the plume traveling out of the model domain." . Don't you condider also the wet removal from incloud and below cloud processes?

– I would suggest to extend the sensitivity analysis including not only the total domain mass but also the maximum traveling range from source for the various bins.

---

## Referee Comment (RC2) · Anonymous Referee #2 · 30 Jan 2020

This paper presents the implementation of a volcanic ash aggregation scheme into the WRF-Chem model. This aggregation scheme is described and applied to simulations of Eyja in 2010. There are two sets of simulations. Sensitivity studies to investigate the properties of the aggregation scheme are presented first, followed by a simulation of the whole Eyja eruption period in 2010. Results from this simulation are compared to in-situ measurements of the ash from DLR and tephra deposits taken over the UK. Overall the paper is well written and clear to follow and provides important evidence that these schemes can be successfully implemented in state-of-the-art VATDs.

General comments: Should aggregation uncertainty be considered in an emergency

response situation? Aggregation will only reduce the distal ash concentrations so maybe computational effort should be put into performing ensemble simulations that vary eruption plume height or the meteorological situation? Can WRF-CHEM be run in real-time emergency response situation? What over head is added by including the representation of aggregation?

Comments on the text:

Ensure that references in the main body of text are in chronological order

L29: Missing "and" between tools, the study of ash physics

L30: Could state what the characteristics of the plume are required for modelling.

L37: Unsure what you mean by plume corner here

L95-102: The text here seems a bit clumsy with section numbers and headings mentioned Would it be possible to include the equations and associated parameters in Table 1 in the main body of text? It would make it easier to follow.

L163: This should be equation 6

L180: This should be equation 7

L181: Refer to Table 3 so the reader knows the particle sizes that the bins refer to.

L205: "radar" missing

L216: You refer to 10km2 as high resolution. This maybe true when considering long range dispersion but is it high enough for modelling aggregation near the eruption plume?

L223: Are 48 hourly meteorological initialisations frequent sufficient?

L244: Why change from 48 hours to 24 hours?

L245: A table outlining the different sensitivity studies would aid the reader here.

L275: How representative is a Df=3.0 of the real world? Does Df vary from volcano to volcano?

L285: Why is the difference between 2.8 and 3.0 highlighted here. Is this unexpected?

L294: Is this jump that is highlighted unexpected?

L306: What are the implications for the different processes being dominant?

L317: The small effect of coupling the aggregation to water emissions seems important. Should this be highlighted more or is it very dependent on the volcano?

L334: 3.9 should be 8.

L334/335: Unsure of the use of "cast", "show" would be clearer.

L345: How much computational expense? Do you have plans to do this?

L346: What does 9C refer to?

L350: Why do you think that there is such a discrepancy between the observations and WRF-CHEM during this time?

L377: 11(d) should be 10(d)

L400: Is wet deposition represented in WRF-CHEM? This can have a large impact on the long-range plume development.

L450: Unsure what is meant by global models here

Comments on Figures

There seems to be a mismatch between using lower case labels on plots and capital letters in the captions. Please make these consistent.

Figure 5: Think about colouring lines to make it easier for the reader to compare lines with same Df.

[Figure]

Figure 6: The subfigure labels are missing. There seems to be a grey bar between the panels. Reorder the legend to make it easier for the reader (e.g. No aggregation all in same column)

Figure 7: Rainbow colour scales are not suitable for people who have colour blindness. Please considering using a different colour scale. Unsure what "Note each time output is at 00hr" means.

Figure 8: As Figure 7 – please consider using a different colour scale.

---

## Author Response (AR1)

[revised manuscript text omitted]

**Commented [SDE3]:** Reviewer one and two both asked that we clarify why we chose a 10 km2 model domain. This is one part of the explanation. The second part is detailed in the conclusions when we suggest that further studies could contain a nested domain over the vent.

**Commented [SDE4]:** We try to explain, here, why the 48 hour model reinitialization was chosen. It was a constraint placed by the availability of computational time. We also reference Hirtl et al (2019) who utilized a similar approach.

**Commented [SDE5]:** Reviewer two requested a table listing the different sensitivity studies included.

eruptive event on May 5th, 2019, initialized at 00:00Z with a rate of 4 x $10^6$ kg s$^{-1}$, which corresponds to an average value of Eyjafjallajökull's largest eruptions. A 72 hour spin up time was included prior to the eruption initialization to allow the meteorological fields to stabilize, and was then run for 6 days, ending 00:00Z on the 11th of May. The smaller model time domain allowed for new meteorological fields to be reinitialized every 24 hours, as opposed to 48 hours in the longer timescale study. Each volcanic ash bin was populated with 10% of the total erupted mass in order to simplify output analysis.

In order to assess how the aggregation code affects model output, WRF-Chem was run with and without the aggregation code enabled. Due to a lack of experimental data, a choice of fractal dimension, $D_f$, is difficult. Therefore, the fractal dimension, $D_f$, was varied to measure its effects on the overall aggregation rate. The span of fractal dimensions chosen ranges from $D_f$ = {2.5, 2.6, 2.7, 2.8, 2.9, 2.95, 2.98, 2.99, 3.0} and is based on studies by Costa et al. (2010) and from a similar study of Mount Saint Helens and Mount Spurr using Fall3D by Folch et al. (2010).

The contribution of each collision kernel, $A_B$, $A_S$ and $A_{DS}$, to the total reduction in domain mass was also assessed by using the same domain and eruption parameters, and enabling only one kernel at a time using a fractal dimension of 2.5 and 3.0. The total change in mass from each kernel was then divided by the total change in mass with all kernels enabled to find the percent contribution.

The impacts of the inclusion of water vapor on the aggregation rate were studied by running the code with and without the 1.8% water vapor emissions included in the model domain. For the simulation run without water vapor emissions, only background water vapor from the FNL datasets were used.

**3.3 Model Setup for April and May 2010 Eruptions of Eyjafjallajökull**

WRF-Chem was also configured to simulate Phase I (April 14-18, 2010) and the Phase III (May 4-18 2010) eruptions of Eyjafjallajökull using the same model domain described above. Phase II eruptions were effusive rather than explosive and ejected tephra at much lower altitudes of 2 to 4 km ASL (Gudmundsson et al., 2012) and were thus not included in this modeling case study.

Eruption source parameters (ESP) for Eyjafjallajökull were adapted from Mastin et al. (2014) and Hirtl et a. (2019). Camera footage and C-band Doppler radar measurements were used to establish three hourly plume heights for the April and May 2010 eruptions (Arason et al., 2011; Mastin et al., 2009; Hirtl et al., 2019). These plume heights were used to calculate eruption rates based on the plume height/eruption rate relationship derived by Mastin et al. (2009). The total erupted mass was then scaled based on work by Gudmundsson et al. (2012) such that the total ash mass ejected over the eruptive phases agreed with the 170 Tg Phase I estimate and 190 Tg Phase III estimates for fine ash stated (Hirtl et al., 2019). The bimodal, silicic (S2) ESP particle size distribution (Table 3) was used to populate the ten volcanic ash bins in the model (Mastin et al., 2009). The three hourly plume heights and eruption rates used in the study are presented in Fig. 2.

In this study, all aggregation collision kernels were enabled, and water vapor emissions as described previously were added to the model domain at each time step. As mentioned earlier, the choice of a fractal dimension is hindered by a

Commented [SDE6]: In reply to reviewer two who asked why we switched to 24 hour reinitializations here.

lack of experimental data. Folch et al. (2010) conducted linear regression analysis of repeated model run comparisons to tephra fallout measurements from eruptions originating at Mount Spurr and Mount Saint Helens. This study resulted in the use of a $D_f$ = 2.99 fractal dimension. Due to a lack of experimental data on the development of volcanic ash fractal dimensions, and the fact that aggregate fractal dimensions are not necessarily constant with time, $D_f$ was set at the upper bound of 3.0, providing a maximum effect of particle aggregation.

**Commented [SDE7]:** Reviewer two asked if a fractal dimension of 3 is realistic. As I mentioned in my reply, it is because 1) Folch et al. note this in the cited work for 2.99 and 2) because in the sensitivity study to follow, a fractal dimension of 2.99 varies only very slightly from 3.0.

**4. Results**

The newly implemented aggregation parameterization was first assessed with a sensitivity study of a singular eruptive event, and then by application to the entire Phase I and Phase III eruption periods.

**4.1. Sensitivity Study Results**

Varying the fractal dimension between 2.5 and 3.0 resulted in a range of aggregation rates. Figure 3 illustrates the change in domain mass from a single 9-hour eruption on May 5th at 00:00Z with a constant eruption rate of 4 x $10^6$ kg s$^{-1}$. As expected, higher values of $D_f$ result in higher rates of aggregation with the largest jumps in the aggregation rate between $D_f$ = 3.0 and 2.8. The degree to which aggregation reduced the overall ash domain mass can be seen in the peak mass loadings at hour 9 in Fig. 3. Here, the peak domain mass using $D_f$ = 3.0 is 17.4 Tg. This is 72% reduction in peak mass compared to the non-aggregation enabled run of 62.9 Tg. Lower values of $D_f$ provide almost no change in the total domain mass. For example, $D_f$ = 2.5 results in a 0.7 % decrease in peak mass by about 0.5 Tg.

To quantify the change in aggregation rate, volcanic ash lifetimes in terms of e-folding were calculated. This analysis is presented in Fig. 4 and indicates a range of e-folding times from 72 hours with no aggregation code enabled to 15 hours with maximum aggregation considered ($D_f$ = 3.0). As the fractal dimension increases, the atmospheric lifetime of volcanic ash decreases due to the incorporation of more volcanic ash particles into each aggregate. When considering fractal dimensions 2.7 and lower, the total lifetime is reduced only slightly, less than 4%. Larger decreases in lifetime become apparent with $D_f$ = 2.8 (10% decrease) and jump thereafter to a maximum 79.5% decrease at $D_f$ = 2.99 and $D_f$ = 3.0 (same decrease for both). Based on work by Folch et al. (2019), it is assumed that an optimal value of the fractal dimension likely lies near $D_f$ = 2.99, which corresponds to a 79.5% difference in e-folding times. In terms of volcanic ash lifetime, on hourly timescales, there is no difference between $D_f$ = 3.0 and 2.99.

**Commented [SDE8]:** Reviewer one requested that we consider a sensitivity study of the extent to which the plume travelled. The study of atmospheric residency time is exactly that, and is presented starting here.

[revised manuscript text omitted]

Commented [SDE9]: We add text here to help Reviewers 1 and 2 understand why this peak was not resolved by WRF. It is a result of the diffusion inherent in gridded moels.

[revised manuscript text omitted]

**Commented [SDE10]:** Reviewers 1 and 2 wanted clarification that WRF does incorporated the effects of water vapor on settling. We've added some text here to clarify this.

**Commented [SDE11]:** Reviewer asked what the implications for the different processes being dominant are. They are mentioned here. Primarily that vertical motion is the primary driver of aggregation.

average 60% of the fallout at two others. This suggests that WRF-Chem may be used to model not only the atmospheric transport of ash clouds, but the deposition of ash as well.

470         Importantly, these observations all suggest that two factors drive volcanic ash aggregation when including aggregation in the WRF-Chem code. First, volcanic ash concentration is noted to be the primary driving factor behind aggregation rate. The majority of model domain mass decreased near the vent where concentrations of ash are high. In addition, PSD analysis indicates that bins with higher portions of the eruption PSD undergo faster rates of ash aggregation. Bins with a larger share of the eruption PSD will aggregate faster due to their increased probability of collision. Second,

475 vertical motions of ash falling through the atmosphere also drive the aggregation process through differential sedimentation for realistic ranges of fractal dimension (between 2.95 and 3.0).

        The inclusion of this aggregation scheme into WRF-Chem provides research and operational meteorological communities a second VATD model to Fall3D that includes volcanic ash aggregation and is the first to run aggregation in an inline fashion where aggregation equations are solved at each model time step (Folch et al., 2010). This inline computation of

480 volcanic ash yields many benefits. For example, the code identifies the driving forces behind volcanic ash aggregation, i.e. ash concentration and differential sedimentation rates, and allows for the study of the effects of water vapor concentration on the aggregation rate. In addition, it allows the study of changes in particle size distributions due to enhanced ash settling as a result of aggregation processes, which are of particular importance to remote sensing communities where the effective particle size directly impacts the spectral methods used for detection. While this study focused primarily on the distal ash cloud transport

485 and aggregation physics, the calculations integrated into WRF would also benefit a higher resolution, nested domain over the emission source to study proximal aggregation effects. The modified code also benefits the operational volcanic ash modeling community by providing model derived ash mass concentrations that augment existing VATD models for use in aircraft hazard mitigation. In the operational setting, first guess, expedient model output from VATD models can be augmented by WRF derived mass loadings as they become available. The time requirement for this is feasible in the operational setting as the

490 modified code is computationally expedient. It ingests output from global models, such as ECMWF and GFS, and runs volcanic ash dispersion and aggregation code while simultaneously calculating mesoscale atmospheric dynamics, eliminating the need for additional, offline calculations. Additionally, this code results in another model that provides researchers a robust treatment of ash microphysical processes as they are erupted, transported and removed from a model domain. Ultimately, this study provides another step towards the inclusion of volcanic ash aggregation, an important physical process, into VATD models.

495 **7. Acknowledgements**

**This publication is the result of research sponsored in part by the NOAA Cooperative Institute for Alaska Research (CIFAR) with funds from NOAA under cooperative agreement NA13OAR4320056 with the University of Alaska Fairbanks (UAF). The Alaska Space Grant Program supported this work. Computational time was provided by UAF Research Computing Systems at the Geophysical Institute and the Department of Defense High Performance**
* * *
**Commented [SDE12]:** This, in addition to earlier comments, helps point out that these equations can be used in studies of the proximal plume using a higher resolution.

**Commented [SDE13]:** The discussion here has been updated to explain why the use of WRF is potentially important to operational forecasting. This stems from a comment by reviewer 2.

[revised manuscript text omitted]

**List of Changes**

765

Line 30: Fixed grammar in reply to reviewer 2.

Line 32: Added a list of plume characteristics as requested by reviewer 2.

770  Line 38: Changed word "edge" to corner as suggested by reviewer 2.

Line 96: Reviewers were asking how or if WRF can be used operationally for aircraft hazard mitigation. We state here that it may be used to augment current VATD models (rather than used alone).

775  Line 145: Added reference to Table 1 as requested by Reviewer 2.

Line 167: Removed bold face font on math terms.

Line 169, 186: Updated equation numbers.

780

Line 187: Added reference to Table 3 as requested by Reviewer 2.

Line 213: Added word "radar" as requested by Reviewer 2.

785  Line 223: Added clarification of the 10km$^2$ resolution as requested by Reviewer 1 and 2 and specified by editor.

Line 233: Added clarification of the choice of 48 hour meteorology field updates, as requested by Reviewers 1 and 2. (We were very limited in our access to computational resources).

790  Line 253: Added new Table 4 as requested by Reviewer 2.

Line 259: Clarified why 24 hour updates were used as requested by both reviewers.

Line 294: Comment on Reviewer 2 questions as to whether or not a fractal dimension of 3.0 is realistic. We argue here that it
795  is.

Line 309: Comment on Reviewer 1 requesting a study on plume travel distance. We argue here that the lifetime analysis does this.

Line 354: Updated figure references as requested by Reviewer 2.

Line 355: Updated "cast" to "show" as requested by Reviewer 2.

Line 368: Added discussion of the peak concentration observed by DLR that was not resovled by WRF as requested by both reviewers.

Line 409: Updated figure labels as requested by Reviewer 2.
Line 433: Both reviewers asked if wet deposition is included in WRF. We had a line in the original text that made it sound like it is not by saying gravitational settling is the only sink – we clarified this here to add that the settling routine does take into account water vapor, so removal is affected by this term.

Line 450: Reviewers questioned what the impact of the different collision kernels on aggregation are. We state this here by noting that vertical motion, correlated to the differential sedimentation kernel, is the driving force behind aggregation.

Line 484: Additional text was added to further clarify that this study was on the distal ash transport, which was questioned by Reviewers 1 and 2 (proximal resolution vs distal resolution).

Line 490: Listed examples of global models for clarification as requested by Reviewer 2.

Figures – Updated all subpanel reverences to match case as requested by Reviewer 2.

Figure 6 – Corrected line that separated both panels to clarify that the legend applies to both as requested by Reviewer 2.

Table 4 – Added new table with summary of sensitivity studies as requested by Reviewer 2.

Replies to Referee One:

Comment 1: "Is the 10x10 km grid size adequate to discribe the near-source aggregates (<15km distance from the volcano)? Could you consider a nested higher resolution grid over Iceland to address these processes? This may not be important for the long range transport since anyhow the bigger particles will be removed from the model but it could provide more insight on the processes and probably improved deposition fluxes near the erruption."

This study built off work by Costa et al. (2010) and Folch et al. (2010, 2015) who used the simplified version of the Smoluchowski equation in this work to study near vent deposition. As such, our efforts focused on the study of ash aggregation processes' effects on distal volcanic ash transport, so attention was paid to the distal plume. The large spatial extent necessary for studying the distal plume required a lower resolution to allow for feasible computational times.

WRF-Chem is capable of much higher resolution model studies and these parameters could be used to study near vent aggregation phenomena, like was done in Costa et al. (2010) and Folch et al. (2010). Furthermore, this study could benefit from a nested domain over the vent, however this was computationally not feasible with the compute time available to our group.

The conclusions section of the paper has been updated to include:

"As stated, the majority of volcanic ash aggregation occurs proximally, especially when high water vapor concentrations are present in the eruptive column. Future studies of volcanic ash near the vent should consider including a nested, high resolution domain over the source to allow for the study of proximal ash fall. We will add a discussion of this to the conclusions portion of the paper in order to highlight the capability of WRF-Chem to include nested, high resolution domains, and add that the equations used also apply to near vent, proximal aggregation."

Comment 2: "Please check that the references are provided in chronological order throughout the text."

All references have been updated chronologically.

Comment 3: "In line 92 " As an example, FALL3D is typically initialized with a WRF model run that is executed prior to the dispersion model. Modeling particle dispersion with WRF-Chem is, therefore, as computationally feasible as running these models since in many cases, a mesoscale, gridded model must be run for their initialization". Indeed, but you can run multiple faster Lagrangian dispersion simulations with different configurations using a single meteorological output (e.g. WRF) which may be important for determining aviation hazard under different emission scenarios."

860

We agree with your comment and will revise the text. Lagrangian dispersion models clearly have their place in aircraft hazard mitigation, especially since they can provide a number of different solutions based on perturbed initial conditions with relatively little computational requirement. The wording of the background has changed to include:

865 "WRF-Chem may augment Lagrangian dispersion models by providing output that is constrained by a number of physical processes, to include aggregation, that are typically not included in dispersion models. Additionally, WRF-Chem may benefit research modeling, allowing researchers to study the effects of numerous microphysical processes on volcanic ash, including aggregation, as well as environmental feedback such as those discussed by Hirtl et al. (2015)."

870 Comment 4: "One peak concentration was observed at 15:30 UTC on April 19, which was not re- solved by WRF-Chem (Fig. 8b). Typical of any Eulerian air quality model, WRF-Chem tends to diffuse ash concentrations, an effect that is also dependent on the model res- olution." I suggest that you should elaborate more on this mismatch between model and observed ash concenntrations. Such high peaks are the primary threat for aviation and moreover these are observed at about 2km elevation which may imply approach or takeoff heights thus increasing the potential danger. This may not be due to Eulerian diffusion

875 otherwise one would expect a more uniform reduction of the concentration fields. Could you please check the concentration at the surounding gridpoints to check if possibly such concentrations exist and are misplaced by the model ?"

We did an analysis of the surrounding grid cells. Laterally there was agreement in the model output with decreased ash seen in the i and j directions. Vertically, however, there was an increase in ash seen aloft, however it was not as extensive as the

880 DLR observed peak. We believe this is Eulerian diffusion since the areas under the curve between the times at this peak agree between the model and observations. The text has been updated to include:

"An analysis of the surrounding grid cells in the vertical and horizontal did not contain this peak, however the next vertical grid cell in the positive k contained higher ash concentrations. This analysis, along with analysis of the integrated volcanic ash

885 over the time span of the peak, lead to the conclusion that this the lack of peak concentration in the model is a result of model diffusion."

Comment 5: "Without aggregation, the only sinks for volcanic ash are via settling or via the plume traveling out of the model domain." . Don't you condider also the wet removal from incloud and below cloud processes?"

890

The volcanic ash settling routing included in WRF-Chem does remove ash faster in the presence of water vapor. It does this by increasing the effective size of the particles, and therefore the fallout rate of ash, with increasing relative humidity. There

is no coupling to rain effects, however, such that rain interactions with volcanic ash are not included. Only the relative humidity fields are taken into account. The text was updated as follows:

"Without aggregation, the only sinks for volcanic ash are via settling, which is dependent on gravity and water vapor concentration, or via the plume traveling out of the model domain."

Comment 6: "I would suggest to extend the sensitivity analysis including not only the total domain mass but also the maximum traveling range from source for the various bins."

Because e-folding time is correlated with distance, a distance sensitivity study would be a recast of this data. The paper may benefit from another figure that details this in terms of distance for each bin, however. We will consider doing so for each bin and if generated will be referenced in the discussion alongside the current e-folding time sensitivity study analysis.

Replies to Referee Two:

Each referee comment is replied to separately:

910 Reply to general comments:

General Comment1: "Should aggregation uncertainty be considered in an emergency response situation?"

As mentioned in the paper, aggregation can reduce the total erupted mass substantially, which will reduce the total atmospheric
915 loading of both proximal and distal ash. For example, we reference Van Eaton et al. (2015), who detailed rapid aggregation of
proximal ash at the onset of the eruption of Mount Redoubt. This reduces the total amount of both proximal and distal volcanic
ash. Aircraft hazard mitigation involves placing limits on the concentration of volcanic ash that commercial aircraft may
encounter. Including volcanic ash aggregation into WRF-Chem, as well as other dispersion models, allows it to capture a more
realistic change in the concentration of ash with time, and therefore more realistic volcanic ash concentrations. Therefore, if
920 an Eulerian model is used in an emergency response, it would benefit from the inclusion of this important microphysical
process. The text has been updated to reflect this by including the following discussion to the background:

"Volcanic ash aircraft hazard mitigation typically focuses on limiting commercial aircraft to ash concentration thresholds
(Casadevall, 1994). WRF-Chem solves the advection equations such that ash concentration is tracked over time. This ability
925 to track volcanic ash mass, rather than particle number, augments current VATD models and offers another tool to constrain
atmospheric ash loading."

General Comment 2: "Aggregation will only reduce the distal ash concentrations so maybe computational effort should be put
into performing ensemble simulations that vary eruption plume height or the meteorological situation."
930

This is a valid approach for modeling volcanic ash dispersion which is already in use. Volcanic ash plume models such as the
aforementioned FPLUME-1.0 detailed by Folch et al. (2015), for example, run a computationally inexpensive set of
calculations that results in parameters which can be input into volcanic ash dispersion models. FPLUME, in addition, includes
parameterizations for volcanic ash aggregation, allowing the forecasting of the resulting particle size distribution in long range
935 deterministic and ensemble models.

While this approach is valid and useful for a number of applications, the integration of volcanic ash aggregation into WRF-
Chem has distinct benefits. First, WRF-Chem can be used to study a number of physical processes involved with the suspension
and transport of volcanic ash in the atmosphere, such as the radiation feedbacks studied by Hirtl et al. (2019) that we mention

940 in the introduction. Including an aggregation option in WRF-Chem allows researchers to include this important microphysical process into the model's treatment of parameterized volcanic ash particle size distributions. Second, volcanic plume models initialize a particle size distribution based on a number of physical processes, to include aggregation. These distributions are then carried forward in the calculations as the proximal plume becomes distal. At this time, the calculations that change the particle size distribution in the distal plume are only based on advection and gravitational settling equations. Aggregation

945 equations allow for another important sink to be considered in the modeling of distal plume ash concentrations.

General Comment 3: "Can WRF-Chem be run in real-time emergency response situation?"

WRF-Chem has not yet been used in an emergency response situation, but it is feasible to consider it for such a purpose. With

950 continued increases in computational power, solving for fully coupled, Eulerian solutions has become increasingly cheap. In our studies, a 4 day simulation with 48 hour spin up time using the model parameters detailed in the paper required less than 20 minutes to complete using 512 processing cores. This could augment current Lagrangian particle dispersion models which are able to provide instant results by providing volcanic ash concentrations which take into account not only gravitational settling and wet deposition, but also aggregation processes.

955

General Comment 4: "What over head is added by including the representation of aggregation?"

The added overhead from the aggregation code is minimal. Because the integration has been reduced to a set of simple algebraic computations, the resulting increase in model time is less than 5%. These effects scale with domain size and a parametric study

960 could be conducted to show the overall increase in overhead with number of cores and domain size.

Text Comment 1: "L29: Missing "and" between tools, the study of ash physics"

These lines have been updated appropriately.

965

Text Comment 2: "L30: Could state what the characteristics of the plume are required for modelling."

These are enumerated later in the text during the model setup, however we updated line 30 to clarify further as follows:

970 "Numerical models have been developed to better describe the initial plume characteristics of eruptions, such as plume height, shape, mass loading and particle size distribution, which are all necessary parameters for ash forecasting. "

Text Comment 3: "L37: Unsure what you mean by plume corner here"

This terminology has been changed to "edge". The term "corner" stemmed from the use of the model grid cell "corner" that acted as the start and end of the distance calculation.

Text Comment 4: "L95-102: The text here seems a bit clumsy with section numbers and headings men- tioned Would it be possible to include the equations and associated parameters in Table 1 in the main body of text? It would make it easier to follow."

The equations in Table 1 are mostly referenced in the text to follow, so we are hesitant to move it farther up, but we will discuss with our editor how to best follow up on this comment.

Text Comment 5 and 6: "L163: This should be equation 6; L180: This should be equation 7"

These equation references have been corrected.

Text Comment 7: "L181: Refer to Table 3 so the reader knows the particle sizes that the bins refer to."

A reference to Table 3 has be added for clarity.

Text Comment 8: "L205: "radar" missing"

This word radar has been added after Doppler.

Text Comment 10: "L216: You refer to 10km2 as high resolution. This maybe true when considering long range dispersion but is it high enough for modelling aggregation near the eruption plume?"

Our study was primarily focused on the dispersion of distal volcanic ash. For a study of near vent volcanic ash fallout, one could use a nested domain with a much higher, for example less than 1 square kilometer, resolution. We now address this in the paper by including the following text:

"As stated, the majority of volcanic ash aggregation occurs proximally, especially when high water vapor concentrations are present in the eruptive column. Future studies of volcanic ash near the vent should consider including a nested, high resolution domain over the source to allow for the study of proximal ash fall. We will add a discussion of this to the conclusions portion

of the paper in order to highlight the capability of WRF-Chem to include nested, high resolution domains, and add that the equations used also apply to near vent, proximal aggregation."

1010 Text Comment 11: "L223: Are 48 hourly meteorological initialisations frequent sufficient?"

These could be more frequent. The choice of 48 hour re-initializations was chosen to offset the very large lag time that was required by the computation cluster in use. Every time WRF was re-initialized the model first had to checkout processors from the cluster. The cluster we used sometimes would queue these jobs for days before launching. Additionally, the 48 hour re-
1015 initialization was used by a study of volcanic ash using WRF-Chem by Hirtl et al. (2019) who observed good results with this interval.

Text Comment 12: "L244: Why change from 48 hours to 24 hours?"

1020 The sensitivity study covered only 6 days which allowed for 24 hour re-initializations. We briefly discuss that this choice is to make the sensitivity study "higher fidelity".

Text Comment 13: "L245: A table outlining the different sensitivity studies would aid the reader here."

1025 The following table has been added to the text to make this more clear.

| Sensitivity Study Analysis Variable | Analysis Method |
| --- | --- |
| Total Domain Mass | Integrate ash mass over entire domain, calculate change in mass over time. |
| Fractal Dimension, $D_f$ | Vary $D_f$ by setting to 2.5, 2.6, 2.7, 2.8, 2.9, 2.95, 2.98, 2.99, 3.0. Analyze change in domain mass using each value. |
| Collision Kernel | Run aggregation code with each collision kernel, $A_B$, $A_S$ and $A_{DS}$ enabled independently. Analyze change in domain mass for each. |

| | |
|---|---|
| Water Vapor Emissions | Ran model with and without enabling water vapor emissions. Analyze change in domain mass for each. |

Text Comment 14: "L275: How representative is a Df=3.0 of the real world? Does Df vary from volcano to volcano?"

This is discussed briefly in the text and will be elaborated upon more. For example, we mention that Folch et al. (2010) detail the correlation between Df and the aggregation rate using an aggregation enabled version of Fall3D. We will expand the discussion with their finding that Df=2.99 was realistic in the cases their study covered. Additionally, the sensitivity study shows little difference in the aggregation rate between Df=3.0 and Df=2.99.

Text Comment 15: "L285: Why is the difference between 2.8 and 3.0 highlighted here. Is this unexpected?"

We are highlighting the change in lifetime seen with varying fractal dimension. Only minimal changes in atmospheric residence time are seen with Df<2.8. The lifetime decreases substantially for Df=2.8 and greater. This is also noted in other studies mentioned in the paper where Df was varied across a range.

Text Comment 16: "L294: Is this jump that is highlighted unexpected?"

It is expected, based on the parametric studies included in Costa et al. (2010).

Text Comment 17: "L306: What are the implications for the different processes being dominant?"

This observation was also noted in Costa et al. (2010) in their parametric study. The main implication is that the contribution from the shear kernel is minimal, and therefore could be disregarded in the calculations.

Text Comment 18: "L317: The small effect of coupling the aggregation to water emissions seems impor- tant. Should this be highlighted more or is it very dependent on the volcano?"

This study focused primarily on the effects of aggregation in the distal plume. Despite the large amount of water vapor emitted from Eyjafjallajökull during its eruptive phases, the overall contribution to atmospheric water vapor as noted from total precipitable water observations during that time were minimal. A study of the effect of water vapor on proximal ash during

the first minutes and hours of the eruption would likely show a greater effect, but the distal plume ends up dry due to the entrainment of dry air by the proximal plume.

Text Comment 19: "L334: 3.9 should be 8."

1060

This has been corrected.

Text Comment 20: "L334/335: Unsure of the use of "cast", "show" would be clearer."

1065 The wording has been changed as suggested.

Text Comment 21: "L345: How much computational expense? Do you have plans to do this?"

Decreasing the grid cell size increases the computational time slightly more than linearly due to the added communication
1070 between compute nodes. This is a known drawback to Eulerian models and we do not plan to resolve these with more higher resolution runs.

Text Comment 22: "L346: What does 9C refer to?"

1075 This should read Figure 8c. This has been corrected.

Text Comment 23: "L350: Why do you think that there is such a discrepancy between the observations and WRF-CHEM during this time?"

1080 We put significant effort into ensuring that this was not an analysis error. The vertical resolution of the model domain is much lower than the horizontal resolution with the most significant spread near the 500mb level. The larger uncertainty in the vertical resulted in differences in the concentration during the transect that were larger than the translational transects, in general. Increasing the vertical resolution of the model increases the computational cost exponentially, as opposed to the near linear increase experienced from increasing horizontal resolution. The text has been updated to include the following discussion:

1085

"An analysis of the surrounding grid cells in the vertical and horizontal did not contain this peak, however the next vertical grid cell in the positive k contained higher ash concentrations. This analysis, along with analysis of the integrated volcanic ash over the time span of the peak, lead to the conclusion that this the lack of peak concentration in the model is a result of model diffusion."

Text Comment 24: "L377: 11(d) should be 10(d):

This has been corrected

Text Comment 25: "L400: Is wet deposition represented in WRF-CHEM? This can have a large impact on the long-range plume development."

The volcanic ash settling routine in WRF-Chem does consider wet deposition by increasing the effective radius of the particles, and thus their fall rate, with increasing relative humidity. The following discussion has been added to make this more clear:

"Without aggregation, the only sinks for volcanic ash are via settling, which is dependent on gravity and water vapor concentration, or via the plume traveling out of the model domain."

Text Comment 26: "L450: Unsure what is meant by global models here"

This is in reference to global spectral models, such as the Global Forecast System run by the National Centers for Environmental Prediction and Integrated Forecast System run by the European Center for Medium-Range Weather Forecasts.

Figure Comment 1: "There seems to be a mismatch between using lower case labels on plots and capital letters in the captions. Please make these consistent."

These figure labels have been updated as suggested.

Figure Comment 2: "Figure 5: Think about colouring lines to make it easier for the reader to compare lines with same Df."

We updated the color schemes with a few different options and will discuss them with the editor to find the best for the final version.

Figure Comment 3: "Figure 6: The subfigure labels are missing. There seems to be a grey bar between the panels. Reorder the legend to make it easier for the reader (e.g. No aggregation all in same column)"

These figures have been corrected to include all labels and are separated by column.

Figure Comment 4: "Figure 7: Rainbow colour scales are not suitable for people who have colour blindness. Please considering using a different colour scale. Unsure what "Note each time output is at 00hr" means."

We will discuss alternative color schemes with our editor that are more easily seen by those with color deficiencies.

Figure Comment 5: "Figure 8: As Figure 7 – please consider using a different colour scale."

We will discuss with the editor our options for different color scales to find the most appropriate for the publication.

**Page 27: [1] Formatted**                 **Unknown**

Font: (Default) +Body (Times New Roman)

**Page 27: [2] Deleted**          **LT Sean D. Egan, USN**          **4/2/20 12:46:00 PM**

... [1]

... [2]

... [3]

... [4]